# The *CXCR4–CXCL12*-Axis Is of Prognostic Relevance in DLBCL and Its Antagonists Exert Pro-Apoptotic Effects In Vitro

**DOI:** 10.3390/ijms20194740

**Published:** 2019-09-24

**Authors:** Katrin Pansy, Julia Feichtinger, Barbara Ehall, Barbara Uhl, Miriam Sedej, David Roula, Beata Pursche, Axel Wolf, Manuel Zoidl, Elisabeth Steinbauer, Verena Gruber, Hildegard T Greinix, Katharina T. Prochazka, Gerhard G. Thallinger, Akos Heinemann, Christine Beham-Schmid, Peter Neumeister, Tanja M. Wrodnigg, Karoline Fechter, Alexander JA. Deutsch

**Affiliations:** 1Division of Hematology, Medical University Graz; Auenbruggerplatz 38, 8036 Graz, Austria; katrin.pansy@medunigraz.at (K.P.); barbara.ehall@medunigraz.at (B.E.); barbara.uhl@medunigraz.at (B.U.); Beata_Prusche@yahoo.de (B.P.); hildegard.greinix@medunigraz.at (H.T.G.); KatharinaTheresa.Prochazka@klinikum-graz.at (K.T.P.); peter.neumeister@medunigraz.at (P.N.); fechterkaroline@gmail.com (K.F.); 2Division of Cell Biology, Histology and Embryology, Gottfried Schatz Research Center for Cell Signaling, Metabolism and Aging, Medical University of Graz, Neue Stiftingtalstraße 6/II, 8010 Graz, Austria; julia.feichtinger@medunigraz.at; 3Otto Loewi Research Center for Vascular Biology, Immunology and Inflammation, Division of Pharmacology, Medical University of Graz, Universitätsplatz 4/I, 8010 Graz, Austria; miriam.sedej@medunigraz.at (M.S.); david.roula@medunigraz.at (D.R.); akos.heinemann@medunigraz.at (A.H.); 4Division of General Otorhinolaryngology, Medical University of Graz, Auenbruggerplatz 26, 8036 Graz, Austria; axel.wolf@klinikum-graz.at; 5Institute of Organic Chemistry, Graz University of Technology, Stremayrgasse 9/4, 8010 Graz, Austria; manuel.zoidl@tugraz.at (M.Z.); t.wrodnigg@tugraz.at (T.M.W.); 6Diagnostic & Research Institute of Pathology, Medical University Graz, Neue Stiftingtalstraße 6, 8010 Graz, Austria; elisabeth.steinbauer@klinikum-graz.at (E.S.); verena.gruber@klinikum-graz.at (V.G.); christine.beham@medunigraz.at (C.B.-S.); 7Institute of Computational Biotechnology, Graz University of Technology, Petersgasse 14/V, 8010 Graz, Austria; gerhard.thallinger@tugraz.at; 8OMICS Center Graz, BioTechMed Graz, Stiftingtalstraße 24, 8010 Graz, Austria

**Keywords:** DLBCL 1, *CXCR4-CXCL12*-axis 2, CXCR4 antagonist 3

## Abstract

In tumor cells of more than 20 different cancer types, the *CXCR4-CXCL12*-axis is involved in multiple key processes including proliferation, survival, migration, invasion, and metastasis. Since data on this axis in diffuse large B cell lymphoma (DLBCL) are inconsistent and limited, we comprehensively studied the *CXCR4-CXCL12*-axis in our DLBCL cohort as well as the effects of CXCR4 antagonists on lymphoma cell lines in vitro. In DLBCL, we observed a 140-fold higher *CXCR4* expression compared to non-neoplastic controls, which was associated with poor clinical outcome. In corresponding bone marrow biopsies, we observed a correlation of *CXCL12* expression and lymphoma infiltration rate as well as a reduction of *CXCR4* expression in remission of bone marrow involvement after treatment. Additionally, we investigated the effects of three CXCR4 antagonists in vitro. Therefore, we used AMD3100 (Plerixafor), AMD070 (Mavorixafor), and WKI, the niacin derivative of AMD070, which we synthesized. WK1 demonstrated stronger pro-apoptotic effects than AMD070 in vitro and induced expression of pro-apoptotic genes of the BCL2-family in CXCR4-positive lymphoma cell lines. Finally, WK1 treatment resulted in the reduced expression of JNK-, ERK1/2- and NF-κB/BCR-target genes. These data indicate that the *CXCR4-CXCL12*-axis impacts the pathogenesis of DLBCL and represents a potential therapeutic target in aggressive lymphomas.

## 1. Introduction

Diffuse large B cell lymphoma is an aggressive lymphoid malignancy and represents the most common subtype of non-Hodgkin lymphoma (NHL) in adults [1]. It arises de novo or by transformation of indolent lymphomas such as follicular lymphomas (FL) [2]. Although DLBCL is in many instances a curable disease, around 40% of patients are refractory or relapse. Based on gene expression profiling, DLBCL can be divided into two different subtypes [3,4]: (i) germinal center B cell-like (GCB-DLBCL), (ii) activated B cell-like (ABC-DLBCL), or in the case of usage of an immunohistochemical algorithm non-germinal center B cell-like (NGCB-DLBCL) [5]. These subtypes are associated with distinctly different overall survival (OS) rates. While GCB-DLBCL patients show favorable overall survival, patients with the NGCB-DLBCL have a worse prognosis [3,4].

The chemokine receptor *CXCR4* and its ligand *CXCL12* are implicated in the retention of B cell precursors and B cell homing to lymph nodes, and therefore play an important role in B cell development [6,7,8,9]. In solid cancer, abnormalities in the *CXCR4–CXCL12*-axis have been linked to many processes including proliferation, survival, migration, invasion, and metastasis [10,11,12], thereby providing evidence for the importance of this chemokine signaling pathway. So far, we and other groups have conducted studies investigating the role of *CXCR4* in DLBCL, whereby the results mainly pointed toward a prominent role of CXCR4 in lymphoma dissemination [10,11,12,13,14,15,16]. However, the data from these studies are to some extent inconsistent and limited, and in particular, combined analyses on *CXCR4, CXCL12*, and *CXCR7*, also known to bind CXCL12 [17], are scarce.

Therefore, we aimed to comprehensively study *CXCR4*, *CXCL12*, and *CXCR7* expression in the DLBCLs samples and corresponding non-neoplastic bone marrow (BM) samples as well as to determine the in vitro effect of two commercially available CXCR4 antagonists, namely AMD3100 and AMD070 [18], and a niacin derivative of AMD070 called WK1, which was generated by us. Hence, we showed that *CXCR4* was higher expressed in DLBCL and that a high *CXCR4* expression was associated with reduced survival. We also demonstrated that *CXCL12* expression correlated to the BM infiltration rate and that *CXCR4* was lower expressed in BM samples from patients exhibiting a remission of lymphoma infiltration after therapy. Both WK1 and AMD070 showed pro-apoptotic effects, which were especially more pronounced in the CXCR4+ lymphoma cell lines treated with WK1. Collectively, our results indicate an impact of the *CXCR4–CXCL12*-axis on lymphomagenesis and its potential role as a therapeutic target.

## 2. Results

### 2.1. High Expression of CXCR4 Is Associated with Poor Clinical Outcome in DLBCL

We determined the expression levels of *CXCR4*, *CXCR7*, and their ligand *CXCL12* [19] in NGCB- and GCB-DLCBLs consisting of primary and transformed follicular lymphomas (*n* = 71 in total) and germinal center B cells (GC-B, *n* = 5) serving as non-neoplastic controls by using RQ-PCR. We observed an average of 140-fold higher *CXCR4* expression in DLBCL and all investigated DLBCL subgroups in comparison to the GC-Bs (Figure 1a, *p* < 0.001), whereas no differential expression was found for *CXCR7* and *CXCL12* (Figure 1a and Appendix A). Furthermore, we observed a 4.7-fold higher *CXCR4* expression in lymphomas with an advanced stage (stage 2–4) compared to DLBCL patients with clinical stage 1 (Figure 1b, *p* = 0.028). BM infiltrating DLBCL displayed a 3.1-fold higher *CXCR4* expression (Figure 1b, *p* = 0.023). Additionally, a positive correlation of *CXCR4* expression and BM infiltration was observed (Spearman rho = 0.550 and *p* < 0.001, Figure 1b). In contrast, no association was found for *CXCL12* and *CXCR7* (Figure 1b and Appendix A).

By dividing the patients into two groups using the third quartile of *CXCR4* mRNA expression, a tendency for an association between high *CXCR4* expression and a poor 5-year-survival rate was observed in our cohort (*p* = 0.088, log-rank test, Figure 1c). Focusing on de novo DLBCL cases, we obtained similar results (*p* = 0.051, log-rank test, Appendix A). This tendency could be confirmed in a public microarray DLBCL dataset [20] (*p* = 0.00018, log-rank test, Figure 1c). For *CXCR7* and *CXCL12* mRNA expression, no association was observed either for the whole lymphoma cohort or for the de novo group (Appendix A).

To determine whether high *CXCR4* and *CXCL12* mRNA expression translated into high protein levels, immunohistochemical analysis for CXCR4 and CXCL12 was performed on the DLBCL samples (Figure 1d, *n* = 40), for which enough material was left. CXCR7 was excluded from further analysis based on its expression profile. For CXCR4 and CXCL12, a significant positive correlation was detected (Spearman rho = 0.714 for CXCR4 and Spearman rho = 0.694 for CXCL12, *p* < 0.01). Additionally, we observed that CXCR4 was exclusively expressed on lymphoma cells (on average 64.5% of lymphoma cells), whereas CXCL12 (on average 30.3% of lymphoma cells) was mainly expressed by lymphoma cells, but was also present in the microenvironment (reactive immune cells and endothelial cells).

### 2.2. CXCR4 Is Somatically Unmutated in DLCBL

Since mutations in the CXCR4 coding sequence frequently occur in B cell lymphomas [21,22], direct sequence analysis was performed on lymphoma samples (*n* = 25) and in lymphoma cells lines (*n* = 4). We could detect a single-nucleotide polymorphism (rs2228014), which had previously been described and derived from publicly available databases [23]. rs2228014, located in exon 2, was found in three of the 25 investigated DLBCL samples and in one of the four investigated cell lines (U2932). Apart from rs2228014, no other alterations were detected (Table 1).

### 2.3. CXCR4-CXCL12-Axis Is Associated with Bone Marrow Infiltration in DLBCL

To further investigate the role of the *CXCR4–CXCL12*-axis in BM infiltration by aggressive lymphomas, we performed RQ-PCR analysis on the corresponding BM biopsies in our lymphoma cohort. In total, 63 BM specimens were used: 52 bone marrow samples were taken at time of diagnosis including 12 patients with BM infiltration at time of diagnosis. From 11 patients, repeated biopsies were taken during their course of disease. Of those, seven patients went into remission while four patients relapsed.

Comparison of *CXCR4* and *CXCL12* mRNA expression levels in BM specimens with and without lymphoma infiltration at time of diagnosis showed a 1.6-fold higher CXCR4 expression in BM specimens exhibiting lymphoma infiltration (Figure 2a, *p* = 0.008). In contrast, no statistically significant difference was detected for *CXCL12* (Figure 2a, *p* = 0.663). However, we observed a strong positive correlation between *CXCL12* expression and percentage of infiltration rate in the investigated BM biopsies (Spearman rho = 0.764, *p* = 0.001).

Furthermore, we analyzed the *CXCR4* and *CXCL12* mRNA expression in seven paired BM samples of previously infiltrated BMs losing infiltration following chemotherapy (BM under remission). Loss of BM infiltration led to a 3.2-fold reduction of *CXCR4* expression (Figure 2b, *p* = 0.032), whereas no significant difference was detected for *CXCL12* (Figure 2b, *p* = 0.382).

Immunohistochemical analysis of CXCR4 and CXCL12 on selected BM specimens (*n* = 19) additionally confirmed the mRNA data. We observed a moderate positive correlation between BM infiltration and protein abundance for both markers (Figure 2c, Spearman rho = 0.595, *p* = 0.031 for CXCR4 and Spearman rho = 0.775, *p* = 0.005 for CXCL12). Interestingly, we detected that in the infiltrated BM samples, an average of 80% of lymphoma cells expressed CXCR4 and an average of 35% of them expressed CXCL12, whereas in the surrounding tissue (stroma) as well as in the BM samples without involvement, less than 30% of the stroma cells expressed CXCR4 and CXCL12 (Figure 2c I–IV).

### 2.4. Treatment of Lymphoma Cell Lines with CXCR4 Antagonists Induced Apoptosis

To investigate the effects of CXCR4-antagonists in vitro, we used the following cell lines: SuDHL4 (as a GCB-DLBCL model), and RI-1 and U2932 (as a NGCB-DLBCL model). Additionally, the lymphoma cell line BL2, which is known to strongly express CXCR4 and to migrate toward CXCL12 in transwell migration assays [24], was included.

First, we characterized the surface expression of CXCR4 in all four investigated cell lines by flow cytometry followed by the CXCL12^AF647^ binding assay combined with blocking antibodies for CXCR4 and CXCR7. CXCR4 expression was found for the BL2, RI-1, SuDHL4, and U2932 cell lines, respectively (Appendix A). Furthermore, we observed that CXCL12AF^647^ was bound just via CXCR4 in the BL2, SuDHL4, and U2932 cells, whereas it was additionally bound via CXCR7 in the RI-1 cells, indicating that these two cell lymphoma lines also express CXCR7 on their surface (Appendix A).

Next, the in vitro effect of the three CXCR4 antagonists (Figure 3a)—AMD3100 (FDA approved), AMD070 and the niacin derivative of AMD070 (termed WK1, synthesized by us)—were investigated. All three antagonists were able to inhibit CXCL12^AF647^ binding in a concentration-dependent manner as demonstrated by the binding assay on BL2 cells (Appendix A). Furthermore, AMD070 inhibited the Transwell migration of BL2 and U2932 cells, while WK1 showed an inhibitory effect only on U2932 cells (Appendix A). Generally, the effects of WK1 were considerably reduced compared to the other two inhibitors in both the above-described assays (Appendix A). Interestingly, we observed reduced growth for AMD070 and WK1 in the BL2 and SuDHL4 cells, respectively, whereas the growth rates of all other investigated cell lines were not affected (Figure 3b). Importantly, AMD3100 and niacin alone did not show any effects (Figure 3b). The IC_50_ values of WK1 were lower compared to those of AMD070 (IC_50_ = 15.4 µM in BL2 and 26.76 µM in SuDHL4 cells for WK1 vs. IC_50_ = 31.18 µM in BL2 and 26.76 µM SuDHL4 cells, Figure 3b). To validate these findings, we treated all four lymphoma cell lines with the three CXCR4 antagonists at concentrations of 1 µM, 5 µM, 10 µM, 20 µM, and 40 µM. We observed that the percentage of Annexin V+ cells in BL2 and SuDHL4 was significantly increased after four treatments with AMD070 and WK1 at 40 µM compared to AMD3100 and DMSO (Figure 3c, *p* < 0.005), indicating the pro-apoptotic effects of both CXCR4 antagonists. Furthermore, the percentage of viable lymphoma cells, Annexin V−/7AAD−, was reduced in BL2 and SuDHL4 when treated with WK1 with concentrations of 10 µM, 20 µM, and 40 µM compared to DMSO and AMD3100 (Appendix A, *p* < 0.01). All other investigated cell lines were unaffected by the three different CXCR4 antagonists. Based on the fact that all of the used lymphoma cell lines overexpressed BCL2 [25,26,27,28] as shown in Appendix A, it seems that the pro-apoptotic effects of AMD070 and WK1 are not influenced by its expression levels.

Finally, we determined the percentage of cells exhibiting cleaved caspase 3 upon DMSO, AMD070, or WK1 treatment of BL2 and SuDHL4 to confirm the pro-apoptotic effects of both antagonists. In both cell lines, two treatments with AMD070 und WK1 resulted in a significantly higher percentage of lymphoma cells staining positive for cleaved caspase 3 (Figure 3d, *p* < 0.01). Remarkably, the percentage of cleaved caspase 3 was significantly higher upon WK1 treatment even at lower concentrations when compared to AMD070 (Figure 3d, *p* < 0.005). Taken together, this suggests that the novel CXCR4–WK1 antagonism leads to strong pro-apoptotic effects on certain lymphoma cell lines.

### 2.5. WK1 and AMD070 Increased Expression of Pro-Apoptotic BCL2-Members

To further dissect the pro-apoptotic effects of the three CXCR4 antagonists, we treated BL2 cells, where apoptosis was induced upon incubation with AMD070 and WK1, respectively, and the U2932 cell line, which was unaffected upon treatment, and determined the gene expression levels of pro- and anti-apoptotic members of the BCL2 family.

In BL2 cells, AMD070 treatment when compared to DMSO resulted in the induction of mRNA expression in two of the eight investigated pro-apoptotic genes (Appendix A I), namely *BAK* (2.4-fold after 1 h, *p* = 0.025) and *NOXA* (3.3-fold after 6 h, *p* = 0.03). Furthermore, two out of three investigated anti-apoptotic genes (Appendix A II), namely *BCL-XL* (4.8-fold after 3 h, *p* = 0.039) and *MCL-1* (3.4-fold after 6 h and 3.7-fold after 12 h, *p* < 0.031) were overexpressed when compared to DMSO. Additionally, AMD070 treatment caused the reduced expression of three pro-apoptotic genes (Appendix A I), namely *BID* (5.2-fold after 6 h, *p* = 0.032), *PUMA* (2.3-fold after 6 h, *p* = 0.042) and the *BIM isoform 9* (3.1-fold after 12 h, *p* = 0.049). In stark contrast, AMD070 treatment of U2932 cells induced expression (at least 2-fold) of five of the eight investigated pro-apoptotic members of BCL2 family members (Appendix A I), namely *BAX* (2.7-fold after 1 h, *p* = 0.014), *NOXA* (5.3-fold after 6 h, *p* = 0.043), *BID* (2-fold after 12 h, *p* = 0.048), *BIK* (6-fold after 6 h, *p* = 0.015), and the *BIM isoform 9* (2.2-fold after 12 h, *p* = 0.046) as well as one of three investigated anti-apoptotic BL2 family members, namely *BCL-XL* (3.7-fold after 1 h, *p* = 0.044, Appendix A II). Furthermore, reduced expression of three pro-apoptotic members (Appendix A I), *BAK* (3.7-fold after 1 h, *p* = 0.018), *BMF* (3-fold after 1 h, *p* = 0.043) and *PUMA* (2.21-fold after 1 h, *p* = 0.035), was observed.

In BL2 cells, the expression levels of seven of the eight investigated pro-apoptotic genes were at least 2-fold induced by WK1 treatment when compared to the DMSO control (*p* < 0.05, Appendix A I), namely, *BAK* (3.4-fold after 1 h, *p* = 0.006), *BIM isoform 9* (2.4-fold after 1 h *p* = 0.011), *BIK* (3.4-fold after 3 h, *p* = 0.041), *BMF* (11.2-fold after 3 h and 3.6-fold after 6 h, *p* < 0.041), *NOXA* (4,2-fold after 3 h and 3.9-fold after 12 h, *p* < 0.041), *BAX* (33.1-fold after 12 h, *p* = 0.029), *PUMA* (2.4-fold, *p* = 0.023), and *BCL-XL* (4.1-fold after 12 h, *p* < 0.0001, Appendix A II) as the anti-apoptotic gene. In contrast to the BL2 cells, WK1 treatment in U2932 cells (Appendix A I and II) caused at least a 2-fold higher expression of *BID* (2.1-fold after 12 h, *p* = 0.042) and *BIK* (7.8-fold after 6 h, *p* = 0.04) as pro-apoptotic genes, a 2-fold lower expression of *MCL-1* (3-fold after 6 h, *p* = 0.04) and *BCL-XL* (4.1-fold after 12 h, *p* = 0.041) as anti-apoptotic genes, and a lower expression of *BAK* (3.5-fold after 1 h, *p* = 0.02) as a pro-apoptotic BCL2 family member.

Comparing WK1 treated BL2 cells to AMD070, five of the eight investigated pro-apoptotic genes (Appendix A I, *p* < 0.043), namely *BIK* (2.1-fold after 3 h and 4-fold after 6 h, *p* < 0.043), *BMF* (2.4-fold after 3 h, *p* = 0.027), *BAX* (33.9-fold after 12 h, *p* = 0.028), the *BIM isoform 9* (3.4-fold after 12 h, *p* = 0.011), *NOXA* (3.8-fold after 12 h, *p* = 0.034), *PUMA* (3.8-fold after 12 h, *p* = 0.025), and *BCL-2* (2.4-fold after 6 h, *p* = 0.044, Appendix A II) as one of the three investigated anti-apoptotic genes were higher expressed in the WK1 treated cells. In contrast, WK1 treated U2932 cells exhibited an at least 2-fold lower expression of four of the eight investigated apoptotic genes (Appendix A I and II), namely *BMF* (2.4-fold after 6 h, *p* = 0.045), *NOXA* (3.4-fold after 6 h, *p* = 0.043), *BCL-XL* (2.4-fold after 12 h, *p* < 0.001), and *MCL-1* (2.4-fold after 6 h, *p* = 0.044).

AMD3100 treatment of BL2 cells (Appendix A II) caused an at least 2-fold increase of the expression of the anti-apoptotic *MCL-1* gene (6.4-fold after 1 h, *p* = 0.009) and reduced expression of two pro-apoptotic BCL2 family members (Appendix A I), namely *BID* (2.4-fold after 6 h, *p* = 0.032) and *PUMA* (2.1-fold after 6 h and 3.3-fold after 12 h, *p* < 0.031). In U2932, AMD3100 treatment caused an at least 2-fold lower expression of three of the eight investigated pro-apoptotic genes (Appendix A I), *p* < 0.031), namely *BAK* (2-fold after 1 h, *p* = 0.031), *BIM isoform 9* (2.2-fold after 1 h, *p* = 0.0286), and *BMF* (4.8-fold after 1 h, *p* < 0.001), and a higher expression of the anti-apoptotic gene (Appendix A II) *BCL-XL* (3.2-fold after 12 h, *p* = 0.008).

### 2.6. WK1 Treatment Causes Downregulation of JNK-, ERK1/2, and NF-κB/ BCR-Targets

To assess whether treatment with CXCR4 antagonists had any effects on three important pathways known to be implicated in lymphomagenesis [29,30,31,32,33], we treated BL2 cells, where apoptosis was induced upon incubation with AMD070 and WK1, respectively, and the U2932 cell line, which was unaffected upon treatment, and determined the mRNA expression levels of target genes of the c-Jun N-terminal kinases (JNK)—(*CCR7, IL-10, CFLAR, ADARB, CCL22* and *FN* (based on the Ingenuity Pathway Analysis tool)), extracellular signal-regulated kinases (1/2) (ERK1/2)—(*cFOS, BUB1, MXD1, JUNB, cJUN, ETV5* and *DUSP1* [34]) and nuclear factor kappa-light-chain-enhancer of activated B cells (NF-κB)/B cell receptor (BCR) pathway (*RGS1, KLF10, TNF, BCL2A1, OAS1,* and *CCL4* [35]) in an explorative manner after 24 h.

In BL2 cells, AMD070 treatment caused a reduced expression of *BUB1* (2-fold, *p* = 0.0115, Appendix A)—one ERK1/2 targets—and *EGR3* (1.6-fold, *p* = 0.0164, Appendix A), a NF-κB/ BCR-target, and a higher expression of *MXD1* (1.6-fold, *p* = 0.0343, Appendix A), a ERK1/2 target, and *RGS1* (1.6-fold, *p* = 0.0343, Appendix A), a NF-κB/ BCR-target. In stark contrast, WK1-treatment caused loss or lower expression of three of six investigated JNK targets (Appendix A), namely the loss of *IL-10* (*p* = 0.021) and lower expression of *CFLAR* (4.8-fold, *p* = 0.0226) and *ADARB* (9.9-fold, *p* = 0.0001), lower expression of four of seven investigated ERK1/2 targets (Appendix A), namely *BUB1* (95.3-fold, *p* = 0.0045), *MXD1* (12.6-fold, *p* = 0.0355), *JUNB* (4.8-fold, *p* = 0.0079), and *DUSP1* (13.7-fold, *p* = 0.0275) and lower or higher expression of five of seven NF-κB/BCR-targets (Appendix A), namely lower expression of *EGR3* (27.9-fold, *p* = 0.0215), *BCL2A1* (5.3-fold, *p* = 0.044), *OAS1* (16.3-fold, *p* = 0.029) and *CCL4* (4.3 fold, *p* = 0.0034) and higher expression of *TNF* (6.4-fold, *p* = 0.042). AMD3100 treatment did not affect the expression levels of any investigated genes. Comparing WK1 treated BL2 cells to AMD070, three ERK1/2 targets, namely *BUB1* (46.5-fold, *p* = 0.001), *MXD1* (20.8-fold, *p* = 0.028), and *DUSP1* (22.8-fold, *p* = 0.038) and four NF-κB/ BCR-targets, namely *RGS1* (9.8-fold, *p* = 0.026), *EGR3* (18.3-fold, *p* = 0.035), *BCL2A1* (8.4-fold, *p* = 0.014), and *CCL4* (6-fold, *p* = 0.003) were lower expressed upon WK1 treatment (Appendix A).

In U2932 cells, AMD070 treatment (Appendix A) caused a lower expression of two ERK1/2 targets, namely *BUB1* (1.7-fold, *p* = 0.048) and *DUSP1* (2.5-fold, *p* = 0.019), and *RGS1* (1.3-fold, *p* = 0.042), a NF-κB/ BCR-target, and higher expression of *MXD1* (1.7-fold, *p* = 0.049). WK1 treatment (Appendix A) caused a higher expression of *FN1* (4.4-fold, *p* = 0.0097) and *ADARB* (3-fold, *p* = 0.008) as the JNK-target, *MXD1* (1.7-fold, *p* = 0.0135) as the ERK1/2 target; and *RGS1* (1.3-fold, *p* = 0.045), *EGR3* (17-fold, *p* = 0.0452), and *TNF* (13.7-fold, *p* = 0.044) as the three NF-κB/BCR-targets. Comparing WK1 treated U2932 cells to AMD070 (Appendix A), higher expression of *IL-10* (1.4-fold, *p* = 0.0132), FN1 (3.7-fold, *p* = 0.0217); two JNK-targets, *MXD1* (1.4-fold, *p* = 0.0095), *cJUN* (2.1-fold, *p* = 0.025), *cFOS* (1.6-fold, *p* = 0,05); and three ERK1/2 targets, *RGS1* (1.7-fold, *p* = 0.0349), *EGR2* (2.1-fold, *p* = 0.025), and *TNF* (3.2-fold, 0.0278) was detected in WK1 treated cells.

## 3. Discussion

This study was designed to comprehensively investigate the expression of the chemokine receptors *CXCR4*, *CXCR7*, and their ligand *CXCL12* [9,17,36] in DLBCL. For all three genes, their implication in cancer surveillance has been demonstrated in more than 20 different solid cancer types [37,38,39]. Their role has also been investigated in DLBCL [10,11,12,13,14,15,16], however, these data are inconsistent and rather limited. Although we used a small number (*n* = 5) of control samples (GC-B cells), we observed *CXCR4* expression in non-neoplastic GC-B cells, but significantly higher expression in DLBCL. By comparing de novo to transformed DLBCL, we did not find any difference, suggesting a similar function of *CXCR4* in both groups. Additionally, we detected higher *CXCR4* expression in DLBCL exhibiting an advanced disease stage. Our data confirmed other studies, which demonstrated decreased *CXCR4* expression for subsets of germinal center B cells and higher expression in DLBCL as well as for patients with advanced-stage disease [10,12,15,40]. However, our analysis revealed that high *CXCR4* expression was not associated with the NGCB-DLBCL subtype, which was indeed observed by Moreno et al. [14], but contradicts previously published data [10]. This discrepancy might be caused by different applied algorithms, which we and others used for NGCB- and GCB-DLBCL classification as well as differences in the DLBCL cohorts. The majority of the included GCB-DLBCL cases in our cohort had been transformed from follicular lymphomas. The distinct genetic and epigenetic alterations of such disease type [41] might, at least partially, explain the observed differences.

In contrast to Moreno et al. [42], who described CXCR7 as a prognostic factor associated with better clinical outcome especially in CXCR4+ DLBCLs, we did not observe any statistically significant association to survival in our cohort. This might be once more caused by differences in the two lymphoma cohorts, since we included a high number of transformed follicular lymphomas in our analysis.

In this study, we observed that CXCL12 is expressed by lymphoma cells and the DLBCL microenvironment. To the best of our knowledge, a comprehensive study on CXCL12 expression in DLCBL has not been performed thus far. Based on our finding that CXCR4 and CXCL12 are expressed simultaneously on lymphoma cells, it might be speculated that an autocrine stimulation loop occurs in aggressive lymphomas.

Survival analysis revealed a trend for high *CXCR4* expression associated with poor survival, in concordance with the findings of two other research groups [13,14,20]. Together with the fact that *CXCR4-CXCL12* signaling activates several pathways like Janus kinases-signal transducer and activator of transcription (JAK-STAT), phosphoinositide 3-kinase (PI3K), protein kinase B (PKB, AKT), mitogen-activated protein kinase (MAPK), and NF-κB [43,44,45], we hypothesize that the high co-occurrence of the receptor and its ligand influences the resistance of lymphomas toward anti-lymphoma therapy. Hence, we expect that low or even no CXCR4 expression leads to none or low activation of the mentioned pathways, thus leading to a higher therapeutic sensitivity of the lymphoma cells. The study of Laursen et al. [13] demonstrated the growth-inhibitory effects of rituximab, a monoclonal antibody, which is used as a standard therapy to treat DLCBL [46], increased by AMD3100, supporting the negative impact of CXCR4 on the chemo-sensitivity of lymphomas cells.

In this study, we observed higher *CXCR4* in DLBCL exhibiting BM infiltration in a third independent DLBCL cohort, in line with previously published results [11,12]. It is known that CXCR4 is upregulated under hypoxic conditions in lymphoma [47] and several other cell lines [48,49,50] in vitro. Since the *CXCR4–CXCL12*-axis and hypoxic conditions have been linked to the BM metastasis of solid cancer [51,52,53] and based on xenograft experiments indicating a role for CXCR4 in bone marrow infiltration of DLBCL [10], it might be possible that this axis is also implicated in lymphoma progression and dissemination. The data on BM samples with or without involvement of DLBCL of our cohort showed a downregulation of *CXCR4* in the BM of patients under remission after therapy as well as the correlation of the *CXCL12* expression with lymphoma cell infiltration underpins this assumption.

Furthermore, we observed that AMD070 and its niacin derivative WK1 had pro-apoptotic effects in the CXCR4+ Burkitt lymphoma and one CXCR4+ GCB-DLBCL cell line as demonstrated by functional assays and gene expression analysis. In contrast, all cell lines (NGCB- and GCB-DLBCL), which were unaffected by the two treatments, were positive for CXCR4 and CXCR7. Moreover, a NGCB-DLBCL cell line, which was also unaffected by both treatments, was positive for CXCR4 only. The observed apoptotic properties of WK1 were even more pronounced compared to AMD070 as demonstrated by the induction of a higher number of pro-apoptotic genes and a higher percentage of apoptotic cells. In stark contrast, AMD3100 and niacin alone had no cytotoxic effects on the lymphoma cell lines. These findings are in line with already published data, where only the inhibition of migration of malignant and non-malignant cells for AMD3100 and growth inhibition for AMD070 have been reported [54,55]. However, our data indicate that the cytotoxic and/or apoptotic effects of AMD070 are increased by the addition of niacin and that these effects depend on the CXCR4 and CXCR7 expression patterns. Based on the gene expression analysis, which demonstrated that WK1 in particular induced more pro-apoptotic genes in a more pronounced manner, it seems that the type and/or subtype of lymphoma cell, especially GCB-DLBCL, may influence the induction of pro-apoptotic genes and response rates. However, based on the fact that AMD070 and WK1 (the two CXCR4 antagonists possessing apoptotic effects) did not up- or downregulate the identical genes, it seems that the underlying molecular mechanism causing cell death is not mediated by the pro-apoptotic BCL2 family members. To the best of our knowledge, BKT140 was identified as the only CXCR4 antagonist possessing high cytotoxic/apoptotic properties in various solid cancer as well as lymphoma and leukemia cell lines so far [24,56,57]. Thus, WK1 represents a second molecule with the same effects as BKT140 and therefore could serve as a starting point for developing new compounds with anti-lymphoma activity.

Our explorative gene expression analysis of JNK-, ERK1/2-, and NF-κB/BCR-target genes demonstrated that three of five JNK targets, four of seven ERK1/2 target genes, and five of seven NF-κB/ BCR-targets were lower expressed in WK1 treated cells, whereas, the effects of AMD070 were diminished and for AMD3100 not detectable. Since JNK-, ERK1/2, and NF-κB/ BCR signaling plays an important role in the development of DLBCL [29,30,31], it could be speculated that the growth inhibitory effects of WK1 might also be mediated by the suppression of these ways.

In conclusion, our data indicate that the *CXCR4–CXCL12*-axis significantly contributes to the pathogenesis of aggressive lymphoma. The underlying mechanisms by which this axis influences the prognostic value may include lymphoma cell growth and dissemination, especially spreading to bone marrow. Based on the observed anti-lymphoma effects of the novel CXCR4 antagonist WK1, which is CXCR4-specific, the *CXCR4–CXCL12*-axis represents an interesting therapeutic target for CXCR4+ GCB-DLCBL at advanced disease stage and with potential BM infiltration.

## 4. Materials and Methods

### 4.1. Patient Samples

Our lymphoma cohort consisted of 71 histologically confirmed DLBCLs including 50 de novo and 21 transformed lymphoma samples (Table 2), receiving a Rituximab containing regimen at the Division of Hematology, Medical University of Graz between 2000 and 2010 (with last follow-up until May 2019). Examined transformed DLBCL samples with an underlying diagnosis of follicular lymphoma only contained the high-grade component. Clonal relationship between an initial FL sample and the resulting transformed DLBCL sample was determined by immunoglobulin heavy chain rearrangement PCR comparing the respectively paired specimens. All samples represented DLBCLs according to the WHO classification [58]. By using the Hans algorithm [5], all cases were classified as follows: 46 cases were categorized as GCB-DLBCL and 25 as NGCB-DLBCL. Determination of the IHC profiles according to the Hans algorithm has previously been described by Fechter et al. [59]. As transformed DLBCL samples originating from FLs exhibited a similar expression pattern as GCB-DLCBL samples [60], these samples were added to this subtype. For this retrospective study, we used patient specimens obtained for routine diagnostic procedures. Hence, no written informed consent of patients was obtained. The study was conducted in accordance with the Declaration of Helsinki, and the protocol was approved by the Ethics Committee of the Medical University Graz (No. 28-516/ex 15/16) on 3 October 2016.

In this study, germinal center B-cells were included as the non-neoplastic control and isolated from tonsils from young patients undergoing routine tonsillectomy as described in detail by our research group [61,62].

### 4.2. Sequencing of CXCR4

Mutational profile by direct DNA sequencing of *CXCR4* was carried out as previously described by our group [63]. In detail, PCR products were purified and sequenced from both sides using the BigDye terminator chemistry 3.1 (Applied Biosystems, Foster City, CA, USA). Sequences were run on an ABI3130-xl automated sequencer (Applied Biosystems, Foster City, CA, USA). For data analysis, the genebank file for *CXCR4* (NG_011587.1) was used. The nucleotide acid sequences for the primer for these purposes are shown in Appendix A.

### 4.3. Cell Lines and Cell Culture

SuDHL4 as a model for GCB-DLBCL, RI-1, and U2932 as a model for NGCB-DLBCL and BL-2 as a model for Burkitt lymphomas, which is known to strongly express CXCR4 and migrate toward CXCL12 [24], were used for the in vitro experiments. SuDHL4, U2932, and RI-1 were cultured in suspension with Roswell Park Memorial Institute (RPMI) 1640 Medium (Gibco, Thermo Fisher Scientific, Waltham, MA, USA) supplemented with 10% heat-inactivated fetal bovine serum (FBS) (Gibco, Thermo Fisher Scientific, Waltham, MA, USA) and 1% Antibiotic-Antimycotic (Gibco, Thermo Fisher Scientific, Waltham, MA, USA). BL-2 cells were maintained in suspension in Roswell Park Memorial Institute (RPMI) 1640 Medium supplemented with 20% FBS (Gibco, Thermo Fisher Scientific, Waltham, MA, USA) and 1% Antibiotic-Antimycotic. Cells were periodically checked for mycoplasma by PCR and were found to be negative. The identity of the DLBCL cell lines was confirmed by variable number tandem repeats (VNTR) analysis using Power Plex 16 System (Promega, Madison, WI, USA), and verified at the online service of the DSMZ cell bank [64]. All cell lines were treated with the commercially available CXCR4 antagonists (MedChemExpress, Sollentuna, Sweden) AMD3100 and AMD070 [18], and the novel niacin derivative of AMD070 called WK1, which was generated by us, in a range from 1 µM to 90 µM. Assays were carried out in triplicate wells and in at least two independent experiments.

### 4.4. Synthesis of WK1

AMD070 (MedChemExpress, Sollentuna, Sweden) (40 mg, 0.12 mmoL, 1 eq) was dissolved in MeOH (800 μL). Nicotinyl chloride hydrochloride (20.4 mg, 0.12 mmoL, 1 eq) and Et3N (32 μL, 0.23 mmol, 2 eq) were added and the reaction mixture was stirred at ambient temperature of 24 h. After consumption of the starting material (detected by TLC: CHCl_3_/MeOH/concd. NH_4_OH = 6/1/0.01, *v*/*v*/*v*) the reaction mixture was concentrated under reduced pressure and purified utilizing silica gel chromatography (CHCl_3_/MeOH/concd. NH_4_OH = 20/1/0.01, *v*/*v*/*v*). WK 1 (20 mg) was obtained as a colorless solid with a yield of 36%. NMR-Spectra, Appendix A).

### 4.5. RNA Extraction and RQ-PCR

Total RNA from fresh frozen DLBCL patient tissues, non-neoplastic germinal centre B cells, and lymphoma cell lines were isolated and cDNA synthesis was performed as previously described by our research group [61,62]. Real-time semi-quantitative PCR (RQ-PCR) for *CXCR4, CXCR7, CXCL12, BAD, PUMA, BAX, BCL-XL, BCL-2, MCL1, BIK, BAK, BIM Iso 9, BID, BMF, NOXA, BCL2A1, CCL4, KLF10, OAS1, RGS1, TNF, EGR3, cFOS, BUB1, MXD1, JUNB, cJUN, ETV5, DUSP1, CCL22, CCR7, IL10, FN1, COL1A, CFLAR*, and *ADARB* (Eurofins Genomic, Ebersberg, Germany and Qiagen, Hilden, Germany; assays and primers are listed in Appendix A) was also performed as previously described [61,62]. *GAPDH, ACTB, PPIA*, and *HPRT1*, known to exhibit the lowest variability among lymphoid malignancies [65], were used as housekeeping genes and relative expression was calculated as described by us [61,62].

### 4.6. Immunohistochemistry

Formalin-fixed, paraffin-embedded tissue was pre-treated in a water bath with Target Retrieval Solution (1:10, Dako, Glostrup, Denmark) for 40 min. Primary antibody to CXCR4 (1:200, order number: ab1640) was purchased from Abcam (Cambridge, UK) and primary antibody to CXCL12 (1:50, order number: MAB350) from R&D Systems (Minneapolis, MN, USA). For staining, kit K5001 (Dako, Glostrup, Denmark) and the automated stainer intelliPATH FLX^®^ (Biocare Medical, Pacheco, CA, USA) were used according to the manufacturer’s instructions. We included tissues known to contain the respective antigens—reactive tonsils—as controls (positive controls). Replacing the primary antibody with normal serum always produced negative results (negative controls). Both negative and positive controls for CXCR4 and CXCL12 are shown in Appendix A. DLBCL specimens were investigated regarding the staining intensities and percentages of the positive stained DLBCL cells according to the following procedure. For determination of the CXCR4 and CXCL12 expression, the whole section was screened for an equal distribution of positive cells. The determination of the percentage was done by calculating the average percentage of cytoplasmic NR4A1 positive cells in at least ten high-power-fields (0.242 mm^2^ each, field diameter: 555.1 µm). Percentages were rounded to 10%.

### 4.7. CXCL12 Binding Assay

We used CXCL12^AF647^ (ALMAC, Craigavon, UK) to determine the binding of CXCL12 to the CXCR4 and CXCR7 positive cells. First, either the CXCR4 antagonists AMD3100, AMD070 or WK1 to a final concentration between 0.01–20 µM or blocking antibodies (10 µg/mL) targeting CXCR4 (clone: 9C4, MBL, Woburn, MA, USA), CXCR7 (clone: 11G8, ChemoCentryx Inc., Mountain View, CA, USA), or isotype controls were added to the cells and incubated for 45 min at 37 °C. The treated cells were further incubated with fluorescent CXCL12^AF647^ (10 ng/mL; BD Biosciences, San Jose, CA USA) for 3 h at 37 °C. Measurement was performed on the LSRII flow cytometer (Becton Dickinson, Franklin Lakes, NJ, USA) using CellQuest analysis software (Becton Dickinson, Franklin Lakes, NJ, USA).

### 4.8. Assessment of Cell Growth

Lymphoma cell lines were plated at a density of 10.000/mL in a 96-well plate and treated with CXCR4 antagonists AMD3100, AMD070, or WK1 in a range from 1 µM to 90 µM. DMSO treated cells and pure medium served as controls and blanks. After treatment, cells were incubated for 72 h at 37 °C and 5% CO_2_. To measure cell proliferation and cytotoxicity, 20 µL EZ4U reagent was added to each well and incubated for 4 h at 37 °C. Results were obtained by absorption measurement at 492 nm with an additional reference measurement at 620 nm using SpectroStar Photometer (BMG LABTECH, Ortenberg, Germany). All experiments were performed in triplicate and repeated at least twice.

### 4.9. Apoptosis Assays

Annexin V/7-AAD Staining: Cells were stained by using Annexin V/7-AAD kit (Biolegend, California, USA). Briefly, 200 µL cell suspension was centrifuged and the supernatant was removed. The pellet was resuspended in 100 µL Annexin V binding buffer (Biolegend, San Diego, CA, USA) and 5 µL Annexin V-APC (BioLegend, San Diego, CA, USA) and 7-AAD (Biolegend, San Diego, CA, USA) were added, followed by incubation for 15 min at room temperature in the dark. Measurement was performed on the LSRII (Becton Dickinson, Franklin Lakes, NJ, USA) flow cytometer using CellQuest analysis software (Becton Dickinson, Franklin Lakes, NJ, USA).

To analyze caspase-3 cleavage, cells were washed and then resuspended in 200 μL 4% paraformaldehyde for 15 min at room temperature in the dark. Cells were permeabilized in methanol and incubated on ice for 30 min. For immunostaining, cells were incubated for 1 h with Cleaved Caspase-3 rabbit mAb -AF647 (Cell Signaling, Cambridge, UK). Measurement was performed on the LSRII flow cytometer (Becton Dickinson, Franklin Lakes, NJ, USA) using CellQuest analysis software (Becton Dickinson, Franklin Lakes, NJ, USA).

### 4.10. Migration Assay

Migration assays were performed using Transwell^®^ inserts (Costar, 6.5 mm diameter, polycarbonate membrane with 5.0 µM pores). Briefly, 3 × 10^5^ cells were resuspended in 100 µL RPMI 1640 medium containing 5 % serum and pre-treated with vehicle or 1 µM of the CXCR4 antagonists AMD70 and WK1 at 37 °C for 2 h. Subsequently, cells were transferred onto the Transwell^®^ inserts and placed into 24-well trays. The lower compartment was filled with 600 µL RPMI 1640 medium with 5% serum-containing 100 ng/mL CXCL12 agonists (AMD070 and WK1) or vehicle. Cells were allowed to migrate for 18 h at 37 °C in a humidified atmosphere and 5% CO_2_. The number of cells migrating to the lower compartment was quantified by flow cytometry. Results are shown as means ± SEMs of *n* = 3–4 independent experiments and are expressed as % of control response.

### 4.11. Microarray Analysis

The E-GEOD-10846 dataset (Affymetrix GeneChip Human Genome U133 Plus 2.0) [20] was download from ArrayExpress and analyzed in R 3.5.1 [66]. By applying rma, the data were preprocessed with the R package ‘oligo’ [67]. Only samples of patients (*n* = 200) who were treated with RCHOP or were assigned a subtype diagnosis were included for further analysis. Expression values for the probe set annotated as CXCR4 were extracted.

### 4.12. Statistical Analysis

For statistical analysis, IBM SPSS Statistics for Windows, Version 23.0 (IBM Corp., New York, USA) was used. *p*-values  <  0.05 were considered statistically significant. The Shapiro–Wilk test was used to test for normality of distribution. Depending on the test result, a t-test or a Mann–Whitney U-test, its non-parametric counterpart, was used to investigate mRNA expression for differences (two-sided *p*-value). 5-year-survival was defined as the time in months from the date of diagnosis to death by any cause.

Survival analysis was performed in R 3.5.1 [66] using the R package ‘survival’ [68] and ‘survminer’ [69]. The patients were split into low- and high-expression groups by using the third quartile of *CXCR4*, *CXCL12* and *CXCR7* expression. Survival was calculated with the Kaplan–Meier method and compared by the log-rank test.

## Figures and Tables

**Figure 1 ijms-20-04740-f001:**
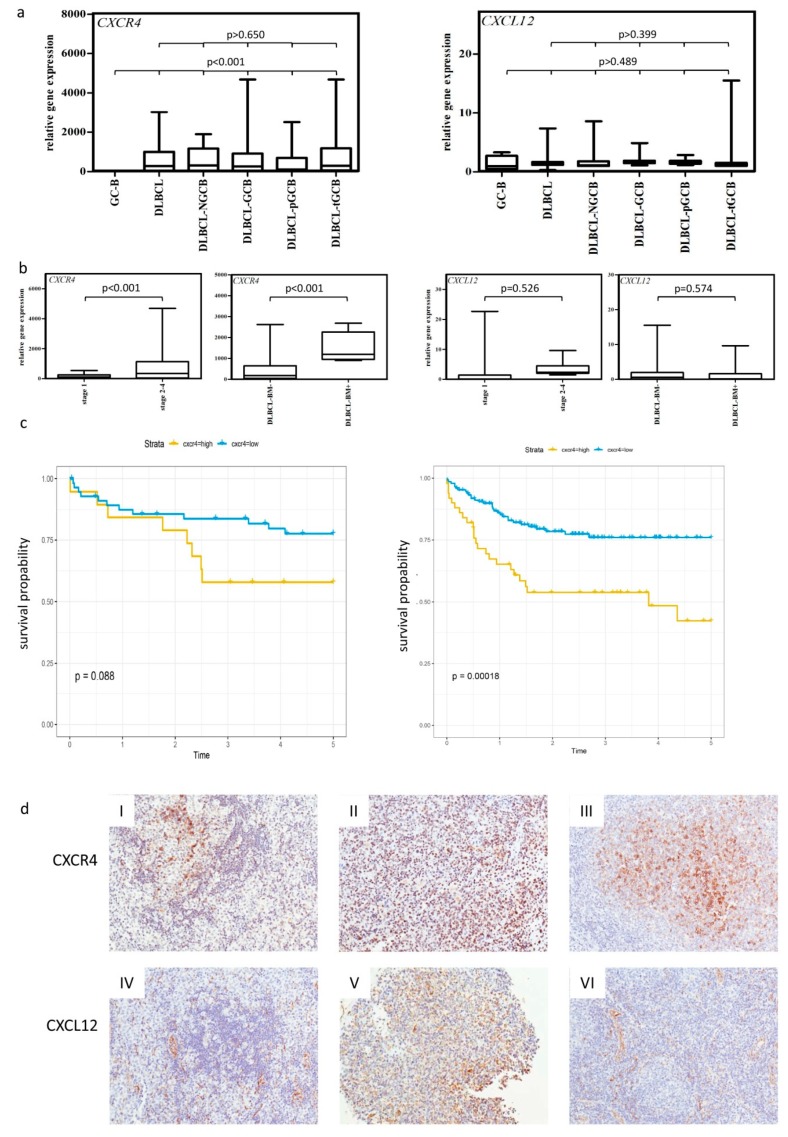
CXCR4 and CXCL12 expression in DLBCL. (**a**) Expression analysis of *CXCR4* and *CXCL12* in non-neoplastic control germinal center B cells (GC-B) and diffuse large B cell lymphoma cells (DLBCL) consisting of DLBCL-NGCB and DLCBL-GCB, by RQ-PCR. GCB-DLBCL were further subdivided into primary (DLBCL-pGCB) and transformed DLBCL (DLBCL-pGCB) originating from follicular lymphoma. (**b**) Expression analysis of *CXCR4* and *CXCL12* in DLBCL samples with early (stage 1) and advanced stage (stage 2–4) (left graphs) and DLBCL samples with and without bone marrow infiltration (right graphs) by RQ-PCR. (**c**) Probability of 5-year-survival in DLBCL patients (our cohort left panel and the cohort of Lenz et al. right [20]) stratified by the third quartile of CXCR4 expression, respectively. (**d**) Representative immunohistochemical stains of CXCR4 (I–III) and CXCL12 (IV–VI) on DLBCL samples (magnification 20×). mRNA expression levels were calculated as a relative expression in comparison to the GC-B cells. All images were captured using an Olympus BX51 microscope and an Olympus E-330 camera.

**Figure 2 ijms-20-04740-f002:**
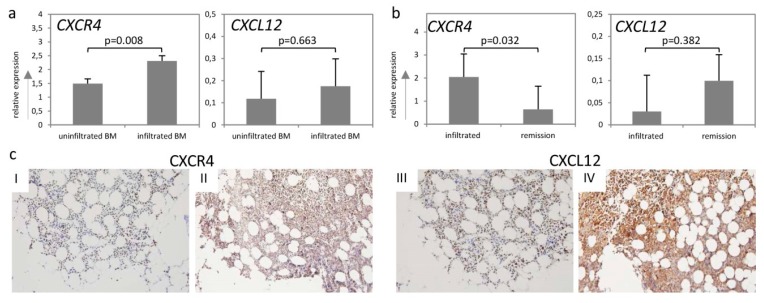
CXCR4 and CXCL12 expression and BM infiltration. (**a**) Expression analysis of *CXCR4* and *CXCL12* in uninfiltrated and infiltrated bone marrow specimens at the time of diagnosis by RQ-PCR. (**b**) Expression analysis of *CXCR4* and *CXCL12* in infiltrated bone marrow biopsies and the respective paired sample in patients under remission by RQ-PCR. (**c**) Representative immunohistochemical stains of CXCR4 (I–II) and CXCL12 (III–IV) on selected bone marrow specimens of DLBCL patients (magnification 20×). (I) and (III) represent the CXCR4 and CXCL12 staining of uninfiltrated bone marrow specimens, and (II) and (IV) represent those of the infiltrated bone marrow specimens. mRNA expression levels were calculated as a relative expression in comparison to uninfiltrated bone marrow specimens. Each bar represents the mean values of expression levels ± standard error of the mean (SEM). The comparison of the expression levels was performed by using the Mann–Whitney U-test or the Student’s t-test. All images were captured using an Olympus BX51 microscope and an Olympus E-330 camera.

**Figure 3 ijms-20-04740-f003:**
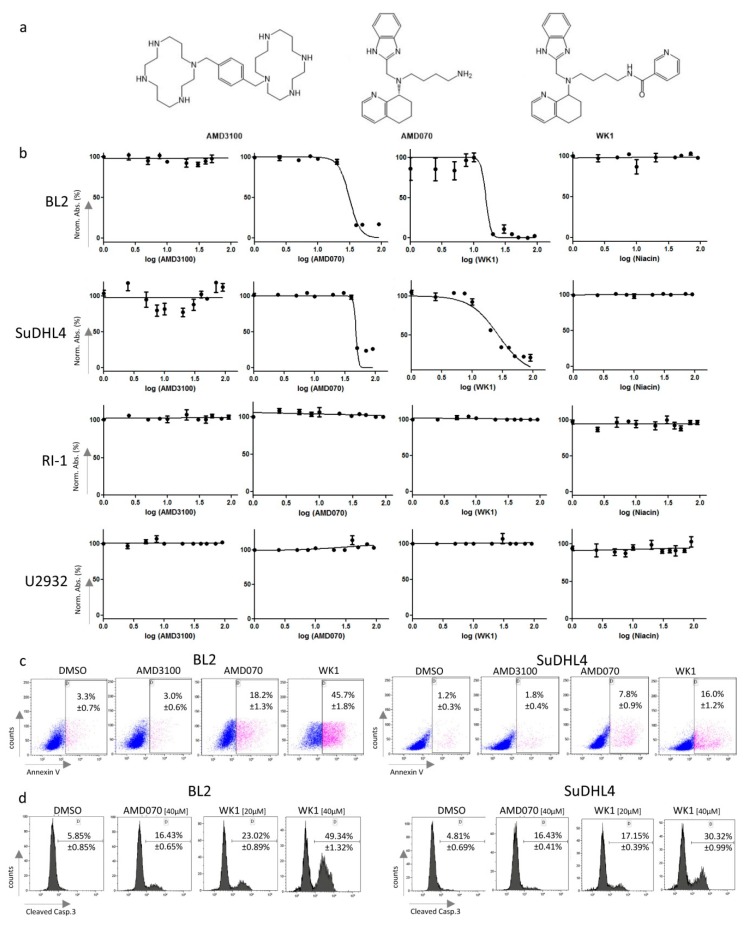
Growth inhibition and apoptosis of B cell lymphoma cell lines upon treatment with CXCR4 antagonists. (**a**) Structure of the CXCR4 antagonists AMD3100, AMD070, and WK1. (**b**) Cell growth of SuDHL4 (as GCB-DLBCL model), RI-1 and U2932 (as NGCB-DLBCL model), and BL2 (as Burkitt model) cell lines in the presence of increasing concentrations (range: 1–90 µM) of the CXCR4 antagonists AMD3100, AMD070, its niacin derivative WK1 and niacin, respectively, as determined by the EZ4U proliferation assay and expressed by percentage of normal absorption. (**c**) Annexin V positivity of BL2 (as Burkitt model) and SuDHL4 (as GCB-DLBCL model) cells treated with AMD3100, AMD070 and its niacin derivative WK1 (concentration: 40 µM; for 48 h) as determined by flow cytometry and compared to the DMSO treated control cells. The treatments and Annexin V staining were performed in triplicate and the medians ± standard deviations are depicted. (**d**) Percentage of cleaved caspase 3 positive BL2 (as Burkitt model) and SuDHL4 (as GCB-DLBCL model) cells treated with 40 µM of AMD070 or 20 µM and 40 µM of its niacin derivative WK1 for 24 h as determined by flow cytometry and compared to the DMSO treated control cells. The treatments and cleaved caspase staining were performed in triplicate and the medians ± standard deviations are depicted.

**Table 1 ijms-20-04740-t001:** Single nucleotide polymorphism occurring in the CDS of CXCR4 in our DLBCL cohort and investigated lymphoma cell lines.

ID	Type		CXCR4 Exon1	CXCR4 Exon2
Al1	ngcb		WT	WT
Al2	gcb	transformed	WT	WT
Al3	gcb	transformed	WT	rs2228014
Al4	gcb	transformed	WT	WT
Al5	ngcb		WT	WT
Al6	gcb		WT	WT
Al8	gcb	transformed	WT	WT
Al9	gcb		WT	WT
Al10	ngcb		WT	WT
Al11	ngcb		WT	rs2228014
Al12	ngcb		WT	WT
Al13	ngcb		WT	WT
Al14	gcb	transformed	WT	WT
Al16	gcb	transformed	WT	WT
Al17	gcb	transformed	WT	WT
Al18	gcb		WT	WT
Al19	gcb	transformed	WT	WT
Al20	gcb	transformed	WT	WT
Al21	ngcb		WT	WT
Al22	gcb		WT	rs2228014
Al26	gcb	transformed	WT	WT
Al28	gcb	transformed	WT	WT
Al33	ngcb		WT	WT
Al34	gcb	transformed	WT	WT
Al71	ngcb		WT	WT
BL2	cell line	Burkitt like	WT	WT
SuDHl4	cell line	GCB like	WT	WT
RI1	cell line	NGCB like	WT	WT
U2932	cell line	NGCB like	WT	rs2228014

WT denotes unmutated.

**Table 2 ijms-20-04740-t002:** Clinico-pathologic parameters of the lymphoma cohort.

Clinico-Pathologic Parameters		Patients (*n* = 71)	Proportion
Subtype	NGCB	25	35.2%
	GCB	46	64.8%
Sex	female	37	52.11%
	male	34	47.9%
Age	≤60a	19	26.8%
	>60a	52	73.2%
Stage	1	16	20.5%
	2	18	23.5%
	3	26	34%
	4	17	22%

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
