# Peer review of "The *CXCR4–CXCL12*-Axis Is of Prognostic Relevance in DLBCL and Its Antagonists Exert Pro-Apoptotic Effects In Vitro"

_ijms, 2019, doi:10.3390/ijms20194740_

Round 1
Reviewer 1 Report
This is an interesting study that can serve as a basis for further research that can lead to recognition of new therapeutic target(s). However, there are some points that need additional consideration.
The most important one is a model for the study. Patients' samples consist of 77 cases of DLBCL that include 51 de novo case and 26 transformed cases. Although transformed cases can be morphologically considered the same entity as de novo ones, molecular underlying mechanisms of development as well as microenvironment of this two groups substantially differ and they should not be considered the same group for molecular analysis and research that evaluates very specific part of lymphomagenesis as in this study. Furthermore, division of de novo DLBCL cases in subtypes should be based on Hans at al. as described in their study and not as authors described in Introduction – adding PBML as a third group. The whole section of Introduction should be checked and information stated in it should be as it was published by original cited literature. Other than Hans algorithm, there are more misinterpretations e. g. reference 8 is about cardiovascular disease and not solid cancers.
Cases that cannot be subtyped because of the lack of the material should not be included in the study.
For DLBCL classification the latest WHO-classification („blue book” from 2017) should be used and not Blood reference listed as 47. Cell line part of the study should include only cell lines that represent DLBCL and not Burkitt lymphoma. Accordingly, discussion and conclusions can then be made for DLBCL (which authors claim was the aim of this study) and not aggressive lymphoma as a broader group (patients' samples were only DLBCL). If the whole study will be done as de novo DLBCL subtyped accordingly to Hans/latest WHO classification (which I strongly recommend) there will be no need for discussion of cohorts as a basis for difference in so far published literature.
In abstract it could be more clearly written which antagonist were used and what is WK1, as well as which „certain“ proapoptotic genes were used.
Whole manuscript should be written with consideration of meaning for words written in italic – Latin phrases such as „in vivo“ should be written in italic. Gene names (or RNAs) should be written in italic as opposed to protein names, based on international convention.
There is no mention of CXCR4 mutation analysis in section Methods and it is not clear if only one polymorphism was analyzed by Q-PCR and specific Taqmans or the gene was sequenced and different mutations were evaluated. Furthermore, only 25 lymphoma samples were used in this part of the study based on CXCR4 expression level. If expression level was the criteria than those with lower expression should be used (possible mutation representing the cause of its lower expression) and gene expression regulation such as a promotor methylation should be analyzed for cases without mutations.
In the section Results it was mentioned that 77 cases were used for gene expression and only 5 control samples – test and control group should be comparable.
For determination of correlation between mRNA and protein expression only 40 samples were used for IHC. There is no explanation why not the whole cohort and why IHC which is semiquantitative method. Western blot could be done on frozen samples.
Author Response
Reviewer 1
Comments and Suggestions for Authors
This is an interesting study that can serve as a basis for further research that can lead to recognition of new therapeutic target(s). However, there are some points that need additional consideration.
Authors: Thank you for the detailed examination of our work, the positive remarks and the suggestions for improvement. Below we address all the points raised and indicate were in the resubmitted manuscript new passages have been introduced.
The most important one is a model for the study. Patients' samples consist of 77 cases of DLBCL that include 51 de novo case and 26 transformed cases. Although transformed cases can be morphologically considered the same entity as de novo ones, molecular underlying mechanisms of development as well as microenvironment of this two groups substantially differ and they should not be considered the same group for molecular analysis and research that evaluates very specific part of lymphomagenesis as in this study. Furthermore, division of de novo DLBCL cases in subtypes should be based on Hans at al. as described in their study and not as authors described in Introduction – adding PBML as a third group. The whole section of Introduction should be checked and information stated in it should be as it was published by original cited literature. Other than Hans algorithm, there are more misinterpretations e. g. reference 8 is about cardiovascular disease and not solid cancers.
Authors: We thank the reviewer for these remarks:
We agree with the reviewer that de novo and transformed DLBCLs possess differences in their microenvironment and their biology. Therefore, we analysed each group separately as clearly demonstrated in the original manuscript. We did not observe any differences in the CXCR4-, CXCL12- and CXCR7 expression levels indicating that all three genes might have a similar function in both groups. Based on this observation and on the fact that in the actual WHO classification both groups belong to the DLBCL entity and that de novo and transformed DLBCLs [1] possess similar expression profiles, we initially decided to perform most of the analysis on the whole cohort. As this issue was raised by the reviewer, we re-evaluated all analysis and added the data obtained from the new analysis on the de novo DLBCL into the supplement figure S2 and the result section 2.1.
Furthermore, we checked the whole introduction section and therefore, we deleted the description on PBML and modified the citation according to the reviewer’s comments (usage of original and right citations).
Cases that cannot be subtyped because of the lack of the material should not be included in the study.
Authors: Based on this comment, we excluded these cases and performed new analysis on 71 cases as depicted in Figure 1 and Figure S1 and described in the results section 2.1. Therefore, we also modified table 1 and the corresponding material section.
For DLBCL classification the latest WHO-classification („blue book” from 2017) should be used and not Blood reference listed as 47. Cell line part of the study should include only cell lines that represent DLBCL and not Burkitt lymphoma. Accordingly, discussion and conclusions can then be made for DLBCL (which authors claim was the aim of this study) and not aggressive lymphoma as a broader group (patients' samples were only DLBCL). If the whole study will be done as de novo DLBCL subtyped accordingly to Hans/latest WHO classification (which I strongly recommend) there will be no need for discussion of cohorts as a basis for difference in so far published literature.
Authors: We want to apologize using the wrong WHO-classification, we corrected everything and use now the right citation.
We included Burkitt lymphoma cell lines for the in vitro testing of CXCR4 antagonists since these models have been used by other studies [2,3], and to use a test system which was described to be suitable for in vitro testing and to make the study more comparable to other studies. In addition, based on our expression analysis demonstrating no differences between de novo and transformed DLBCL, we are quite convinced that CXCR4 might possess a similar function in these two subgroups of the lymphoma entity and potentially to other lymphoma entities at least as a therapeutic target. That is the reason, we kept the expression and survival analysis in the modified version. Based on that, we tried to broaden the conclusion.
In abstract it could be more clearly written which antagonist were used and what is WK1, as well as which „certain“ proapoptotic genes were used.
Authors: The abstract has been modified according to this comment.
Whole manuscript should be written with consideration of meaning for words written in italic – Latin phrases such as „in vivo“ should be written in italic. Gene names (or RNAs) should be written in italic as opposed to protein names, based on international convention.
Authors: We have checked and have modified all Latin phrases and gene names based on the international convention to italic notation as demanded by the reviewer.
There is no mention of CXCR4 mutation analysis in section Methods and it is not clear if only one polymorphism was analyzed by Q-PCR and specific Taqmans or the gene was sequenced and different mutations were evaluated. Furthermore, only 25 lymphoma samples were used in this part of the study based on CXCR4 expression level. If expression level was the criteria than those with lower expression should be used (possible mutation representing the cause of its lower expression) and gene expression regulation such as a promotor methylation should be analyzed for cases without mutations.
Authors: We want to apologise that we have forgotten to provide information on the mutation analysis of CXCR4. We now implemented the used method –direct sequencing in the Material and Methods section. This analysis was done because it was reported that some B cell malignancies possess CXCR4 mutation [4,5]. Since CXCR4 was significantly up-regulated and high CXCR4 expression was detected in all investigated sample, we do not believe that we would observe any promotor methylation.
In the section Results it was mentioned that 77 cases were used for gene expression and only 5 control samples – test and control group should be comparable.
Authors: The recommended non-neoplastic controls for DLBCL are germinal centre B cells isolated from reactive tonsils of healthy donors by cell sorting. Based on the fact that it is very difficult to obtain unfixed tonsils, it was impossible for us to increase the number of patients.
For determination of correlation between mRNA and protein expression only 40 samples were used for IHC. There is no explanation why not the whole cohort and why IHC which is semiquantitative method. Western blot could be done on frozen samples.
Authors: Because the whole cohort has already being used for several studies [6–12], we lack material and therefore IHC analysis was performed on only 40 samples.
The reasons why we performed IHC instead of Western blot, was also the limited amount of frozen tissue and the fact that it is possible to evaluate by which cell the investigated genes are expressed by this method. This is very important for chemokine and chemokine receptors since these are frequently expressed by reactive immune cells in the microenvironment.
Reviewer 2 Report
Minor comments
It could be useful to characterize the panel of lymphoma cell line for the expression of Bcl2 It could be useful to have more clinical informations relative to the cohort of lymphoma patients (if available) if available to add in Table 2. It would be interesting to perform cell biology experiments on CXCR4+CXCR7+ cell lines with siRNA for CXCR4 and/or CXCR7 and on silenced cell lines to test the compounds (AMD070 and WK1) used in the manuscript. Please check and revise in the text in vitro (in vitro) Please introduce in the Abstract section what is WK1
Author Response
Minor comments
It could be useful to characterize the panel of lymphoma cell line for the expression of Bcl2.
It could be useful to have more clinical informations relative to the cohort of lymphoma patients (if available) if available to add in Table 2.
It would be interesting to perform cell biology experiments on CXCR4+CXCR7+ cell lines with siRNA for CXCR4 and/or CXCR7 and on silenced cell lines to test the compounds (AMD070 and WK1) used in the manuscript.
Please check and revise in the text in vitro (in vitro) Please introduce in the Abstract section what is WK1
Authors: Thank you for carefully reading our work and for the raised issues. Below we address all points raised and indicate the changes in the manuscript accordingly.
Concerning the BCL2 expression status of the used lymphoma cell lines, we implemented its status into the supplementary Table S1 based on our examination of literature and correlated this data to the pro-apoptotic effects of the used CXCR4 antagonists.
In our point of view, we included the most important clinical data in table 2.
Performing silencing experiments of CXCR4 and/or CXCR7 followed by CXCR4 antagonist treatment would be very interesting. But it was impossible for us to perform such an experiment within 17 days (time of revsion).
We have checked and revised all the italic notation of “in vitro” throughout the whole manuscript.
Finally, we have introduced the description of WK1 in the abstract.
Reviewer 3 Report
In this manuscript by Pansy et al., the authors investigate whether expression of the chemokine receptor CXCR4 in a cohort of patients with diffuse large B cell lymphoma (DLBCL) served as an indicator of a poor patient outcome. They also examine the CXCR4 ligand, CXCL12, as well as the CXCR7 receptor, which also binds CXCL12. Consistent with previous reports, they found that CXCR4 expression appeared to be elevated in patient samples, and higher expression was associated with decreased survival and involvement of bone marrow. They did not observe elevated expression of CXCR7 or CXCL12 in patient samples. The authors also examined the effects of three different CXCR4 antagonists, included one that they developed, WK1, on survival of transformed B-cell lines. They found that their molecule was more effective in inducing apoptosis than the other agonists, and therefore warrants further development. This is an interesting study that corroborates previously published reports and provides the initial characterization of a CXCR4 inhibitor that could potentially be developed as a therapeutic agent against DLBCL and other maladies. The following comments and suggestions might help improve the quality of the manuscript.
In the Introduction, reference 6 is not relevant/correct for the general role of CXCR4 in B-cell homing and development.
In Figure 1, data for the A. and B. graphs would more be appropriately shown as box-whisker plots or a scatter of data points since third quartile CXCR4 expression levels are examined for survival probability in C. Also placing DLBCL-GCB in the same graph with DLBCL-pGCB and DLBCL-tGCB may be misleading, as these latter groups appear to be subsets of DLBCL-GCB. The meaning of p- and tGCB should be stated in the figure legend. More importantly, there is little information explaining the immunohistochemistry imaged in D. Positive and negative controls are mentions in the Materials and Methods, but are not shown. If CXCR4 plays an important role in tumor cell survival, why is there no difference in its expression between primary and transformed samples? Perhaps the authors should speculate on this question in the Discussion. Also for the Discussion and related to data in Figure 1, is high CXCR4 in the DLBCL samples related to rituximab treatment (Laursen 2019 Oncotarget)?
In Figures S2, was the CXCR4 antibody labeled with PE or was a labeled, secondary antibody used? IT=isotype control.
In Figures 4 and 5, no clear trend immerged from the gene expression results that correlated with the apoptotic phenotype. Therefore, the underlying molecular events leading to cell death by AMD070 and WK1 in BL2 cells were not revealed by these data. Perhaps these figures should be included as supplementary data. Why did the authors not examine protein phosphorylation rather than mRNA levels in Figure 5?
The amount of each antagonist used in Figures 3-5 is high (see Figure S3). Perhaps the apoptotic effects are CXCR4-independent. Because AMD070 has a higher affinity for CXCR4 than AMD3100 (Murakami et al. 2009 Antimicrob Agents Chemother), perhaps similar apoptotic effects would occur with higher AMD3100 concentrations.
In the Materials and Methods, information regarding CXCR4 sequencing appears to be missing. Based on the text, it is assumed that the authors only detected one of the five known variants of CXCR4. The information regarding how immunohistochemistry images were quantified appears to be missing. The number of replicates for gene expression analyses appears to be missing.
In the Discussion the authors state that in their analysis CXCR7 was not associated with a better clinical outcome. However, they did not show a survival probability plot comparing the first and third quartile levels if CXCR7 expression as they did with CXCR4. These data could be included as supplementary data.
Author Response
Comments and Suggestions for Authors
In this manuscript by Pansy et al., the authors investigate whether expression of the chemokine receptor CXCR4 in a cohort of patients with diffuse large B cell lymphoma (DLBCL) served as an indicator of a poor patient outcome. They also examine the CXCR4 ligand, CXCL12, as well as the CXCR7 receptor, which also binds CXCL12. Consistent with previous reports, they found that CXCR4 expression appeared to be elevated in patient samples, and higher expression was associated with decreased survival and involvement of bone marrow. They did not observe elevated expression of CXCR7 or CXCL12 in patient samples. The authors also examined the effects of three different CXCR4 antagonists, included one that they developed, WK1, on survival of transformed B-cell lines. They found that their molecule was more effective in inducing apoptosis than the other agonists, and therefore warrants further development. This is an interesting study that corroborates previously published reports and provides the initial characterization of a CXCR4 inhibitor that could potentially be developed as a therapeutic agent against DLBCL and other maladies. The following comments and suggestions might help improve the quality of the manuscript.
Authors: We thank the editor for the detailed examination of our submitted paper and for pointing out the strengths of our study. All raised issues are addressed below and all modifications are highlighted in the resubmitted manuscript.
In the Introduction, reference 6 is not relevant/correct for the general role of CXCR4 in B-cell homing and development.
Authors: It has been modified.
In Figure 1, data for the A. and B. graphs would more be appropriately shown as box-whisker plots or a scatter of data points since third quartile CXCR4 expression levels are examined for survival probability in C. Also placing DLBCL-GCB in the same graph with DLBCL-pGCB and DLBCL-tGCB may be misleading, as these latter groups appear to be subsets of DLBCL-GCB. The meaning of p- and tGCB should be stated in the figure legend. More importantly, there is little information explaining the immunohistochemistry imaged in D. Positive and negative controls are mentions in the Materials and Methods, but are not shown. If CXCR4 plays an important role in tumor cell survival, why is there no difference in its expression between primary and transformed samples? Perhaps the authors should speculate on this question in the Discussion. Also for the Discussion and related to data in Figure 1, is high CXCR4 in the DLBCL samples related to rituximab treatment (Laursen 2019 Oncotarget)?
Authors: We thank the reviewer for this comment. We modified Figure 1 and supplementary Figure S1: The expression data is now depicted by using box-whisker plots. Furthermore, we stated the meaning of DLBCL-pGCB and DLBCL-tGCB in both figure legends.
Furthermore, a new supplementary Figure S9 showing negative and positive control has been added into the supplementary data file.
Based on the fact that we detected no difference between primary and transformed DLBCL, we speculated that CXCR4, CXCL12, and CXCR7 possess a similar molecular function in both subgroups. This speculation was now implemented in the discussion section.
Besides, we added and commented on the study of Laursen et al in the Discussion section.
In Figures S2, was the CXCR4 antibody labeled with PE or was a labeled, secondary antibody used? IT=isotype control.
Authors: We stated that we used a PE-labeled CXCR4 antibody and explained what IT means in the figure legend.
In Figures 4 and 5, no clear trend immerged from the gene expression results that correlated with the apoptotic phenotype. Therefore, the underlying molecular events leading to cell death by AMD070 and WK1 in BL2 cells were not revealed by these data. Perhaps these figures should be included as supplementary data. Why did the authors not examine protein phosphorylation rather than mRNA levels in Figure 5?
Authors: We disagree with the reviewer concerning the expression data depicted in Figure 4 and 5. They clearly demonstrate differences between the two inhibitors and the two cell lines and help to understand their molecular function. Based on this and the fact nothing was stated by the two other reviewers, we believe that these two figures should stay main figures.
Concerning the reason, why we present mRNA data instead of phosphorylation: These RQ-PCR assays were perfectly established and available in our lab.
The amount of each antagonist used in Figures 3-5 is high (see Figure S3). Perhaps the apoptotic effects are CXCR4-independent. Because AMD070 has a higher affinity for CXCR4 than AMD3100 (Murakami et al. 2009 Antimicrob Agents Chemother), perhaps similar apoptotic effects would occur with higher AMD3100 concentrations.
Authors: We thank the reviewer for this important issue. The concentrations of the used CXCR4 antagonist are quite high. However, based on the fact that the cell growth effects depend on the CXCR4 and/or CXCR7 expression, we are convinced that the apoptotic effects are CXCR4 dependent.
Based on the fact that we used concentration up to 100µM and these concentrations are quite high, we are convinced that AMD3100 does not possess any growth inhibitory/apoptotic effects.
In the Materials and Methods, information regarding CXCR4 sequencing appears to be missing. Based on the text, it is assumed that the authors only detected one of the five known variants of CXCR4. The information regarding how immunohistochemistry images were quantified appears to be missing. The number of replicates for gene expression analyses appears to be missing.1
Authors: We thank the reviewer for this important comment. We implemented information on CXCR4 mutation and IHC analysis in the Material and Methods section.
In the Discussion the authors state that in their analysis CXCR7 was not associated with a better clinical outcome. However, they did not show a survival probability plot comparing the first and third quartile levels if CXCR7 expression as they did with CXCR4. These data could be included as supplementary data.
Authors: Based on this important comment, we implemented CXCR7 survival plots into the suppl. Figure S2 (d & e).
References
Davies, A.J.; Rosenwald, A.; Wright, G.; Lee, A.; Last, K.W.; Weisenburger, D.D.; Chan, W.C.; Delabie, J.; Braziel, R.M.; Campo, E.; et al. Transformation of follicular lymphoma to diffuse large B-cell lymphoma proceeds by distinct oncogenic mechanisms. Br. J. Haematol. 2007, 136, 286–293, doi:10.1111/j.1365-2141.2006.06439.x.
Beider, K.; Ribakovsky, E.; Abraham, M.; Wald, H.; Weiss, L.; Rosenberg, E.; Galun, E.; Avigdor, A.; Eizenberg, O.; Peled, A.; et al. Targeting the CD20 and CXCR4 pathways in non-hodgkin lymphoma with rituximab and high-affinity CXCR4 antagonist BKT140. Clin. Cancer Res. 2013, 19, 3495–3507, doi:10.1158/1078-0432.CCR-12-3015.
Wester, H.J.; Keller, U.; Schottelius, M.; Beer, A.; Philipp-Abbrederis, K.; Hoffmann, F.; Šimeček, J.; Gerngross, C.; Lassmann, M.; Herrmann, K.; et al. Disclosing the CXCR4 Expression in Lymphoproliferative Diseases by Targeted Molecular Imaging. Theranostics 2015, 5, 618–630, doi:10.7150/thno.11251.
Treon, S.P.; Cao, Y.; Xu, L.; Yang, G.; Liu, X.; Hunter, Z.R. Somatic mutations in MYD88 and CXCR4 are determinants of clinical presentation and overall survival in Waldenstrom macroglobulinemia. Blood 2014, 123, 2791–2796, doi:10.1182/blood-2014-01-550905.
Hunter, Z.R.; Xu, L.; Yang, G.; Zhou, Y.; Liu, X.; Cao, Y.; Manning, R.J.; Tripsas, C.; Patterson, C.J.; Sheehy, P.; et al. The genomic landscape of Waldenstrom macroglobulinemia is characterized by highly recurring MYD88 and WHIM-like CXCR4 mutations, and small somatic deletions associated with B-cell lymphomagenesis. Blood 2014, 123, 1637–1646, doi:10.1182/blood-2013-09-525808.
Deutsch, A.J.A.; Rinner, B.; Pichler, M.; Prochazka, K.; Pansy, K.; Bischof, M.; Fechter, K.; Hatzl, S.; Feichtinger, J.; Wenzl, K.; et al. NR4A3 Suppresses Lymphomagenesis through Induction of Proapoptotic Genes. Cancer Res. 2017, 77, 2375–2386, doi:10.1158/0008-5472.CAN-16-2320.
Unterluggauer, J.J.; Prochazka, K.; Tomazic, P.V.; Huber, H.J.; Seeboeck, R.; Fechter, K.; Steinbauer, E.; Gruber, V.; Feichtinger, J.; Pichler, M.; et al. Expression profile of translation initiation factor eIF2B5 in diffuse large B-cell lymphoma and its correlation to clinical outcome. Blood Cancer Journal 2018, 8, 79, doi:10.1038/s41408-018-0112-5.
Wenzl, K.; Hofer, S.; Troppan, K.; Lassnig, M.; Steinbauer, E.; Wiltgen, M.; Zulus, B.; Renner, W.; Beham-Schmid, C.; Neumeister, P.; et al. Higher incidence of the SNP Met 788 Ile in the coding region of A20 in diffuse large B cell lymphomas. Tumor Biology 2016, 37, 4785–4789, doi:10.1007/s13277-015-4322-1.
Troppan, K.; Wenzl, K.; Pichler, M.; Pursche, B.; Schwarzenbacher, D.; Feichtinger, J.; Thallinger, G.G.; Beham-Schmid, C.; Neumeister, P.; Deutsch, A. miR-199a and miR-497 Are Associated with Better Overall Survival due to Increased Chemosensitivity in Diffuse Large B-Cell Lymphoma Patients. Int. J. Mol. Sci. 2015, 16, 18077–18095, doi:10.3390/ijms160818077.
Deutsch, A.J.A.; Rinner, B.; Wenzl, K.; Pichler, M.; Troppan, K.; Steinbauer, E.; Schwarzenbacher, D.; Reitter, S.; Feichtinger, J.; Tierling, S.; et al. NR4A1-mediated apoptosis suppresses lymphomagenesis and is associated with a favorable cancer-specific survival in patients with aggressive B-cell lymphomas. Blood 2014, 123, 2367–2377, doi:10.1182/blood-2013-08-518878.
Prochazka, K.T.; Posch, F.; Deutsch, A.; Beham-Schmid, C.; Stöger, H.; Abdyli, L.; Greinix, H.; Pichler, M.; Neumeister, P. Immunohistochemical double hit score enhances NCCN-IPI and is associated with detrimental outcomes in refractory or relapsing patients with diffuse large B cell lymphoma. Br J Haematol 2018, 183, 142–146, doi:10.1111/bjh.14912.
Fechter, K.; Feichtinger, J.; Prochazka, K.; Unterluggauer, J.J.; Pansy, K.; Steinbauer, E.; Pichler, M.; Haybaeck, J.; Prokesch, A.; Greinix, H.T.; et al. Cytoplasmic location of NR4A1 in aggressive lymphomas is associated with a favourable cancer specific survival. Sci. Rep. 2018, 8, doi:10.1038/s41598-018-32972-4.
Round 2
Reviewer 1 Report
The manuscript is now greatly improved, but still some issues about the model remain:
although 5 control samples are very small group it can be accepted, but should be mentioned in discussion and results explained considering this fact IHC staining performed on 40 samples due to the lack of the material should be stated clearly and results obtained viewed from that perspective or study can be updated with material from other DLBCL cases that were not used in previous studies Burkitt lymphoma cell lines results/experiments should be omitted from the study
Author Response
Reviewer 1
The manuscript is now greatly improved, but still some issues about the model remain:
although 5 control samples are very small group it can be accepted, but should be mentioned in discussion and results explained considering this fact IHC staining performed on 40 samples due to the lack of the material should be stated clearly and results obtained viewed from that perspective or study can be updated with material from other DLBCL cases that were not used in previous studies Burkitt lymphoma cell lines results/experiments should be omitted from the study
Authors: We thank the reviewer for the detailed examination and pointing out the improvements of the work
Based on his/her comments, we modified the manuscript, which is highlighted in the submitted version:
We now mentioned the small size of our control group in the discussion section. Furthermore, in the IHC results, we added now the reason, why we did not examine the whole cohort.
Finally, we omitted the results of the Raji cell line since it is a Burkitt lymphoma cell lines. We adapted the whole in vitro part.
In our point of view the data of the second Burkitt lymphoma cell line -BL2- is important for the manuscript because it was described to be a cell line that strongly expresses CXCR4 and migrates towards CXCL12 in vitro [1]. In our study, it is one of three cell line that just expresses CXCR4 but not CXCR7. CXCR4 expression is lower in the two cell lines (RI-1 and U2932, Fig S3a), in which WK1 and AMD070 showed the same effects indicating that CXCR4 expression levels influence the in vitro effects of CXCR4 antagonist (Fig 3b).
Reviewer 3 Report
Most of the corrections and explanations were satisfactory.
However, the authors disagree with the comment that no clear trend in gene expression emerged from the graphs shown in Figures 4 and 5 that correlate with the apoptotic phenotype. In the rebuttal, the authors state that the results clearly demonstrate differences between the two inhibitors and the two cell lines. I still disagree with the authors on this point. For BL2 cells, AMD070 and WK1 both induced cell death at the concentrations used for the gene expression analysis. Therefore, one would expect that genes affected by the drug treatment to be either up-regulated by both drug treatments or down-regulated by both. Given that U2932 cells were insensitive to these drugs, those genes affected similarly by both drugs in BL2 cells would not be expected to be affected in U2932 cells. The authors should state whether any genes they examined fulfill these two criteria. Time points may not be important for this analysis. For example, NOXA is upregulated at 1hr in BL2 cells and at 5hr in U2932 cells; nonetheless this gene is upregulated in both cell lines. Therefore, this gene would not fulfill the two criteria.
Author Response
Most of the corrections and explanations were satisfactory.
However, the authors disagree with the comment that no clear trend in gene expression emerged from the graphs shown in Figures 4 and 5 that correlate with the apoptotic phenotype. In the rebuttal, the authors state that the results clearly demonstrate differences between the two inhibitors and the two cell lines. I still disagree with the authors on this point. For BL2 cells, AMD070 and WK1 both induced cell death at the concentrations used for the gene expression analysis. Therefore, one would expect that genes affected by the drug treatment to be either up-regulated by both drug treatments or down-regulated by both. Given that U2932 cells were insensitive to these drugs, those genes affected similarly by both drugs in BL2 cells would not be expected to be affected in U2932 cells. The authors should state whether any genes they examined fulfill these two criteria. Time points may not be important for this analysis. For example, NOXA is upregulated at 1hr in BL2 cells and at 5hr in U2932 cells; nonetheless this gene is upregulated in both cell lines. Therefore, this gene would not fulfill the two criteria.
Authors: We also want to thank the reviewer for carefully reading our work and for the raised issue:
We agree with him that none of the analysed genes in Figure 4 and 5 (in the new version: Supplementary figure S7 and S9) fulfil the criteria that these were up- or down-regulated by both antagonists indicating that the underlying molecular mechanism could not be identified by this gene expression analysis. So, we decided, as suggested by the reviewer in the first review, to include the figures in the supplement.
References
[1] K. Beider et al., “Targeting the CD20 and CXCR4 pathways in non-hodgkin lymphoma with rituximab and high-affinity CXCR4 antagonist BKT140,” (eng), Clinical cancer research : an official journal of the American Association for Cancer Research, vol. 19, no. 13, pp. 3495–3507, 2013.